# Natural emotion vocabularies as windows on distress and well-being

Vera Vine [1✉], Ryan L. Boyd [2] & James W. Pennebaker [3]

To date we know little about natural emotion word repertoires, and whether or how they are associated with emotional functioning. Principles from linguistics suggest that the richness or diversity of individuals' actively used emotion vocabularies may correspond with their typical emotion experiences. The current investigation measures active emotion vocabularies in participant-generated natural speech and examined their relationships to individual differences in mood, personality, and physical and emotional well-being. Study 1 analyzes stream-of-consciousness essays by 1,567 college students. Study 2 analyzes public blogs written by over 35,000 individuals. The studies yield consistent findings that emotion vocabulary richness corresponds broadly with experience. Larger negative emotion vocabularies correlate with more psychological distress and poorer physical health. Larger positive emotion vocabularies correlate with higher well-being and better physical health. Findings support theories linking language use and development with lived experience and may have future clinical implications pending further research.

[1] University of Pittsburgh, Pittsburgh, PA, USA. [2] Lancaster University, Lancaster, UK. [3] University of Texas at Austin, Austin, TX, USA. ✉email: vinevj@upmc.edu

In today's age of hyper-self-awareness, the ability to name emotions is often celebrated. It is often assumed that people who use rich emotional vocabularies are emotionally and physically healthier than those who express themselves using a narrower range of emotion words. Self-styled emotion experts publish lengthy lists of emotion words to help people articulate feelings as precisely as possible[1–4]. In popular and scholarly press, it is proposed that naming emotions can promote mental and physical health[5–7]. To capitalize on this effect, readers are advised to "beef up your emotion concepts" and "learn as many new words as possible," to be equipped to categorize difficult emotions when they arise more flexibly and precisely[5]. Mobile applications ensure that the most precise emotion label is only a finger-click away[8].

Despite this interest in naming emotions, we still know little about natural emotion word repertoires, and whether or how natural emotion vocabularies (EVs) are associated with emotional functioning. Most research on benefits of identifying emotions has measured self-perception of emotional abilities instead of emotion language itself[9,10]. The studies more concerned with emotion language have relied on passive presentation of experimenter-generated emotion words, capturing constructs other than natural EVs. For example, emotional intelligence, the ability to recognize and reason using emotions, is measured using multiple choice formats[11]. Emotion differentiation, the ability to distinguish same-valenced emotions conceptually, which is correlated with positive mental health, is inferred from the structure of Likert emotion ratings[12]. Faster responding on such Likert-type emotion scales is also associated with helpful emotion regulation[13]. And compared to viewing unlabeled images, viewing upsetting images paired with a matching emotion word activates frontal lobe structures that dampen emotional intensity[14]. Such studies point compellingly to benefits related to recognizing common emotion concepts and labels, but they have unclear relevance to natural, spontaneous emotion word use in everyday life.

Linguistic approaches contribute a useful distinction to the study of emotion language: degrees of verbal knowledge fall into two nested domains. An active vocabulary is the set of words an individual produces spontaneously, which constitutes only a subset of one's passive vocabulary, or the full body of words the person can recognize[15–17]. Importantly, sizes of active and passive vocabularies are not correlated; passive vocabularies increase through schooling, whereas active vocabularies tend to plateau, suggesting that people reuse the words with which they are most comfortable[16]. Studies presenting participants with emotion labels reveal processes involving passive emotion knowledge. However, as others have pointed out[12,18–20], to fully understand the role of emotion language in well-being, we must also extend research into active EVs.

At their most basic level, words are symbols that correspond to concepts and experiences[21]. From this perspective, there should be a broad alignment between active vocabularies and experience. At this stage, we are agnostic about the causality in this relationship, which could be bidirectional. Specifically, active vocabularies could correspond with experiences for at least three, non-mutually exclusive reasons. First, active vocabularies provide a window into mental habits. According to Zipf's[22] principle of least effort, speakers are naturally economical in their use of language, with active vocabularies driven by utility. Like carpenters who keep their most useful tools within arm's reach, speakers use most frequently the words that perform their most common mental operations. This linguistic principle has become a central premise of personality research: active vocabularies can tell us about the concepts people use in their thinking most[22–25]. By this logic, an individual may simply have developed a wider variety of labels for certain emotions via more frequent experiences of them.

Second, active vocabularies may reflect expertise or interest. The psychologist uses a rich vocabulary of psychology words, the sommelier a rich vocabulary of wine words. Lévi-Strauss[26] famously recorded that indigenous hunter-gatherers in the Philippines easily named over 450 plants, 75 birds, and 20 varieties of ants. Lévi-Strauss reasoned that utility alone could not explain such staggering vocabularies, as there would be diminishing practical returns on such fine-grained classifications. Instead, he speculated that interest may have motivated these exceedingly diverse taxonomies. In addition to well-established determinants of vocabulary acquisition and maintenance[27,28], we similarly suggest that preoccupation with or interest in one's own affective states could contribute to the development of increasingly diverse affective taxonomies and lexica.

Third, it appears that experience can grow into gradual alignment with words. The strong causal view—that language fully determines experience[29]—has been dismissed[30], but subtler versions of this hypothesis are compelling. Dewey[31] has described the function of words as "a fence, a label, and a vehicle—all in one"[31], meaning that words not only divide our continuous stream of experiences into discrete units, but also catalog experiences in memory for future use, and conceptually scaffold our interpretations of future events. Several others have articulated similar roles for language in shaping experience (see language-as-context[32]; the mangrove effect[33]; essence place-holders[34]). Initial experiments seem to confirm that verbal concepts help construct perceptions of reality, including the experience and interpretation of emotional states[5,34–37].

In the present project, two studies examine the characteristics of active EVs and their relationships to individual differences in mood, personality, and physical and emotional well-being. We expect a broad cross-sectional correspondence between words and experience, such that large vocabularies for negative emotions would signal low well-being, while large positive EVs would signal high well-being.

## Results

**Study 1**. Stream-of-consciousness writing, with its unstructured nature, presents an ideal opportunity to investigate linguistic markers of individual differences[24]. Study 1 investigated EVs in stream-of-consciousness writings (final $N = 1567$) by (1) identifying basic properties of positive and negative EVs, including their size and test–retest reliability; (2) examining the link between EVs and broad individual differences in demographic characteristics, personality, and physical and emotional health; and (3) examining the relationships between EVs for specific emotion families (i.e., anger, sadness, anxiety/fear, and happiness) and the intensity of corresponding state-level moods. Relationships are expressed with standardized errors and 95% bias-corrected and accelerated confidence intervals, generated using 2000 bootstrapped replicates with replacement.

Overall, 6.11% ($SD = 1.66$) of words used in essays were emotionally toned, based on the positive emotion ($M = 3.62$, $SD = 1.28$) and negative emotion ($M = 2.40$, $SD = 1.15$) categories computed using linguistic inquiry and word count (LIWC)[38]. The actual number of unique emotion words was far smaller. Average EV was 0.55 for negative emotions ($SD = 0.36$, range: 0–5.71, 95% CI: 0.53–0.57]), and 0.52 for positive emotions ($SD = 0.34$, range: 0–3.75, 95% CI: 0.29–0.34]). Based on the EV algorithm, these average rates correspond to approximately one unique positive and one unique negative emotion word per 200 words of text. Test–retest correlations were modest ($r_{NegEV} = 0.18$, 95% CI: 0.10–0.27], $p < 0.001$; $r_{PosEV} = 0.28$, 95% CI: 0.19–0.38], $p < 0.001$),

**Table 1 Pearson and partial correlations of emotion vocabulary (EV) with other study variables for study 1 ($N = 1567$ unless marked otherwise).**

| | Pearson correlations with negative EV | Partial correlations with negative EV | Pearson correlations with positive EV | Partial correlations with positive EV |
|---|---|---|---|---|
| Negative EV | – | – | – | – |
| Positive EV | 0.16 (0.06)*** | 0.18 (0.04)*** | – | – |
| Demographic variables | | | | |
| Age | −0.02 (0.02) | −0.05 (0.03)† | −0.06 (0.02)* | −0.04 (0.02) |
| Gender[a] | 0.20 (0.02)*** | 0.15 (0.03)*** | −0.00 (0.03) | 0.06 (0.03)* |
| Individual differences—text-derived | | | | |
| Cognitive processing | 0.08 (0.03)** | – | −0.07 (0.03)** | – |
| Negative emotional tone | 0.61 (0.04)*** | – | 0.01 (0.06) | – |
| Positive emotional tone | −0.03 (0.04) | – | 0.50 (0.03)*** | – |
| General vocabulary size | 0.11 (0.04)*** | – | 0.21 (0.03)*** | – |
| Illness words | 0.11 (0.05)*** | 0.01 (0.03) | 0.06 (0.04)* | 0.06 (0.03)* |
| I-words | 0.25 (0.06)*** | 0.23 (0.04)*** | 0.05 (0.04) | 0.10 (0.04)*** |
| We-words | −0.11 (0.02)*** | −0.06 (0.02)* | −0.05 (0.02)* | −0.06 (0.03)* |
| Affiliation words | 0.05 (0.03)† | 0.09 (0.03)*** | 0.14 (0.04)*** | 0.06 (0.03)* |
| Achievement words | 0.06 (0.06)* | 0.04 (0.04)† | 0.11 (0.04)*** | 0.00 (0.03) |
| Leisure words | −0.07 (0.06)** | −0.02 (0.04) | 0.20 (0.06)*** | 0.08 (0.05)** |
| Individual differences—self-reported | | | | |
| Openness[b] | −0.03 (0.03) | −0.02 (0.03) | 0.04 (0.03) | 0.01 (0.03) |
| Conscientiousness[b] | −0.01 (0.03) | 0.06 (0.03)* | 0.06 (0.03)* | 0.07 (0.03)* |
| Extraversion[b] | −0.04 (0.03) | −0.03 (0.03) | 0.06 (0.03)* | 0.03 (0.03) |
| Agreeableness[b] | 0.01 (0.03) | 0.05 (0.03)† | 0.09 (0.03)** | 0.06 (0.03)† |
| Neuroticism[b] | 0.17 (0.03)*** | 0.08 (0.03)** | −0.09 (0.03)** | −0.02 (0.03) |
| Depression symptoms[c] | 0.11 (0.03)*** | −0.01 (0.03) | −0.07 (0.03)* | −0.01 (0.03) |
| Overall health[d] | −0.13 (0.03)*** | −0.05 (0.03)± | 0.06 (0.03)* | 0.05 (0.03) |

*Note*: Partial correlations control for general vocabulary, negative, and positive emotional tone. All tests are two-tailed. Coefficients are expressed as **r** (SE). For 95% confidence intervals and exact significance values, see Supplementary Table 5.
[a]Coded 0 = male, 1 = female.
[b]$n = 1341$ participants based on available data.
[c]$n = 1256$ participants based on available data.
[d]$n = 1545$ participants based on available data.
***$p < 0.001$
**$p < 0.01$
*$p < 0.05$
†$p < 0.10$

but in line with previous findings of temporal stability for traits manifested in verbal behavior ($r = 0.24$)[39]. Positive and negative EV indices were modestly correlated with each other in a positive direction, suggesting that to some extent they may reflect a unitary tendency toward greater diversity in emotion language (Table 1). As shown in Table 1, negative EV was associated with female gender, while positive EV was not. Neither negative nor positive EV were related to age.

Indicating convergent validity, negative and positive EV were related to cognitive processing words (see Table 1). Both positive and negative EV were also associated with general vocabulary, and each index was associated with the corresponding emotional tone. These convergences support our general conceptualization of EV; they are also further relevant to our examination of incremental validity, below.

As seen in Table 1, negative EV was generally associated with prevalence of linguistic markers related to poor well-being, namely, lower frequency of we—words and leisure words, and higher use of I—words and illness words. Positive EV was generally associated with markers related to positive well-being: high frequency of achievement, affiliation, and leisure words. A few findings were not expected. Positive EV was unexpectedly related to higher mention of physical illness words. Achievement words did not show the expected inverse correlation with negative EV, perhaps because these concerns are common to most students in a university setting.

As shown in Table 1, negative EV was related to higher neuroticism and depression and lower overall health. Conversely,

positive EV corresponded with indicators of more positive experiences and higher psychosocial functioning: higher extraversion, agreeableness, and overall health, and lower self-reported neuroticism and depression. For scatterplots of key relationships see Supplementary Fig. 2. Although not the focus of these analyses, Study 1 data were also used to examine criterion validity of the text-derived indices. Correlations between text-derived and self-reported indicators of well-being, reported in Supplementary Table 2, consistently indicate the associations suggestive of validity (i.e., I-words and illness words with low well-being; we-words, affiliation, achievement, and leisure words with high well-being).

Given the possibility that EV is partly a product of emotional tone of texts and/or individuals' general verbal ability, the analyses reported in Table 1 were repeated using partial correlations controlling for general vocabulary, negative emotional tone, and positive emotional tone. As indicated by partial correlations in Table 1, many key relationships between emotion EV and psychological variables remained or became significant. Thus, EV appears to be capable of explaining unique variance in health and adjustment indices, above and beyond the effects of overall verbal development and emotional tone. Readers interested in an exploration of the interaction between negative and positive EV are directed to Supplementary Note 1 and Supplementary Fig. 1.

Students used more diverse negative EV when they felt negatively before writing ($r = 0.19$, SE = 0.02, 95% CI: 0.14–0.24], $p < 0.001$), and larger negative EV was also related

**Table 2 Partial correlations between emotion vocabulary (EV) for distinct emotion types and changes in self-rated moods in study 1 ($N = 1546$).**

| Emotion vocabulary | Sadness mood change | Worry mood change | Anger mood change | Stressed mood change | Positive mood change |
|---|---|---|---|---|---|
| Sadness | 0.09 (0.03)*** | 0.02 (0.03) | −0.08 (0.02)** | 0.01 (0.03) | −0.01 (0.03) |
| Fear | −0.03 (0.02) | 0.09 (0.03)*** | −0.12 (0.02)*** | 0.02 (0.03) | 0.06 (0.03)* |
| Anger | 0.01 (0.03) | 0.05 (0.03)† | 0.10 (0.03)*** | 0.05 (0.03)† | −0.10 (0.03)*** |
| Undifferentiated negative | 0.00 (0.03) | 0.06 (0.03)* | −0.01 (0.03) | 0.09 (0.03)** | −0.02 (0.03) |
| Positive | −0.04 (0.02) | 0.02 (0.03) | −0.07 (0.02)** | −0.02 (0.03) | 0.04 (0.02)† |

*Note:* Values are partial correlation coefficients between EV indices and post-writing ratings of subjective mood. Each correlation controls for pre-writing levels of the target mood, as well as general vocabulary, and negative and positive emotional tone. Sample size is based on availability of state mood ratings. All tests were two-tailed. Coefficients are expressed as **r** (SE). For 95% confidence intervals and exact significance values, see Supplementary Table 6.
***$p < 0.001$.
**$p < 0.01$.
*$p < 0.05$.
†$p < 0.10$.

to feeling negatively after writing ($r = 0.21$, SE = 0.02, 95% CI: 0.17–0.26], $p < 0.001$). Similarly, the positive EV and was related to positive self-reported mood before writing ($r = 0.19$, SE = 0.02, 95% CI: 0.14–0.23], $p < 0.001$). and after writing ($r = 0.22$, SE = 0.02, 95% CI: 0.18–0.27], $p < 0.001$).

Emotion-specific EV scores were used to examine the relationship between variability in emotion language with change in corresponding mood states. Sadness vocabularies were used to predict post-writing levels of self-reported sadness, fear vocabularies to predict post-writing worry, anger vocabularies to predict post-writing anger, and undifferentiated negative vocabularies to predict self-reported levels of post-writing stress. To provide a stringent test of the effects of sheer emotion vocabulary size, apart from overall vocabulary richness or the emotionality of the writing, partial correlations controlled for general vocabulary and negative and positive emotional tone. To isolate change in self-reported moods over time, each partial correlation also controlled for pre-writing levels of the target mood. As shown in Table 2, as a function of variability in specific emotion vocabularies, the corresponding subjective feelings grew stronger over the course of writing, and these subjective mood effects were highly specific to the target mood. People who used more names for sadness grew sadder over the course of the stream of consciousness exercise, but did not grow more worried, angry, or stressed. People who used more names for fear grew more worried, but did not feel sadder, angrier, or more stressed. People who used more names for anger in their writing grew angrier, but actually grew less worried, and reported no change in stress. The people who grew more stressed over the course of the writing exercise were those who used high rates of unique undifferentiated negative words. Positive EV showed a similar correspondence with increases in positive mood.

**Study 2**. Study 2 analyzed a large collection of public blogs (final $N = 35, 385$). Bloggers wrote often over several years, producing text samples spanning an unrestricted range of topics—some personal and emotional, others dry and factual. Relationships in Study 2 are expressed with standardized errors and 95% bias-corrected and accelerated confidence intervals, generated using 500 bootstrapped replicates with replacement.

The average blogger used approximately 6.55 unique negative emotion words and 5.99 unique positive emotion words. The negative EV rate averaged 0.29 (SD = 0.21, range: 0–2.66), and the positive-EV rate averaged 0.33 (SD = 0.21, range: 0–2.49), or just less than one unique positive and one unique negative emotion word per 300 words of text. To assess EV stability, each

blog was split in half, and separate EV statistics were computed for each half. Reliabilities ($r_{NegEV} = 0.27$, SE = 0.01, 95% CI: 0.26–0.29], $p < 0.001$; $r_{PosEV} = 0.28$, SE = 0.01, 95% CI: 0.27–0.29], $p < 0.001$) exceeded both the test–retest reliability in Study 1 and rates found previously for psychological linguistic variables[39]. As in Study 1, positive and negative EV were positively correlated ($r = 0.22$, SE = 0.01, 95% CI: 0.20–0.23], $p < 0.001$). Examples of emotion words captured appear in Supplementary Methods.

As in Study 1, negative EV was associated with female gender; positive EV was also associated with female gender (see Table 3). Replicating Study 1, negative and positive EV were related to cognitive processing words. Both positive and negative EV were associated with general vocabulary, and each index was associated with the corresponding emotional tone.

Negative EV was again associated with linguistic markers of low well-being. As in Study 1, negative EV was correlated with higher use of illness and *I*-words, and lower use of *we*- and leisure words. Exceeding effects of Study 1, negative EV was further related to lower rates of achievement words. Results involving positive EV were similar to Study 1. Positive EV was again related to higher rates of achievement, leisure, and affiliation words. Unexpectedly, correlations with illness and I-words, which had been positive but nonsignificant in Study 1, were larger and reaching significance in Study 2. For scatterplots of key relationships see Supplementary Fig. 2.

Almost all relationships between negative emotion EV and psychological variables remained significant after controlling for other factors (and in the same direction), as did several of the relationships involving positive emotion EV. In other words, EV again appeared to be capable of explaining unique variance in health and adjustment indices.

## Discussion

These studies examined whether emotion vocabularies in natural language are associated with psychosocial functioning. Study 1 indicated that EV indices are psychometrically acceptable. Regarding construct validity, EVs were correlated with cognitive processing tendency, general vocabulary, and emotional tone in logically consistent directions. Most markers of well-being were associated with use of sparser negative and more expansive positive EVs. At the state level, using a wider array of emotion words was associated with intensification of the corresponding mood. These effects were strikingly emotion-specific: people with varied vocabularies for sadness grew sadder, people with varied anger vocabularies grew angrier, and so on—even when

**Table 3 Pearson correlations of emotion vocabulary (EV) with other study variables for study 2 ($N = 35,385$).**

| | Pearson correlations with negative EV | Partial correlations with negative EV | Pearson correlations with positive EV | Partial correlations with positive EV |
|---|---|---|---|---|
| Negative EV | – | – | – | – |
| Positive EV | 0.22 (0.01)*** | 0.12 (0.01)*** | – | – |
| Demographic variables | | | | |
| Age[a] | −0.09 (0.01)*** | 0.01 (0.01) | 0.05, (0.01)*** | −0.07 (0.01)*** |
| Gender[b] | 0.15 (0.01)*** | 0.15 (0.01)*** | 0.07 (0.01)*** | 0.07 (0.01)*** |
| Individual differences—text-derived | | | | |
| Cognitive processing | 0.21 (0.01)*** | – | 0.08 (0.01)*** | – |
| Negative emotional tone | 0.51 (0.01)*** | – | −0.03 (0.01)*** | – |
| Positive emotional tone | 0.09 (0.01)*** | – | 0.35 (0.01)*** | – |
| General vocabulary size | 0.24 (0.01)*** | – | 0.46 (0.00)*** | – |
| Illness words | 0.16 (0.01)*** | 0.07 (0.01)*** | 0.07 (0.01)*** | 0.06 (0.01)*** |
| I-words | 0.28 (0.01)*** | 0.20 (0.01)*** | 0.13 (0.01)*** | 0.10 (0.01)*** |
| We-words | −0.08 (0.01)*** | 0.00 (0.01) | −0.02 (0.01)** | 0.00 (01) |
| Affiliation words | −0.01 (0.01) | 0.06 (0.01)*** | 0.08 (0.01)*** | 0.03 (0.01)*** |
| Achievement words | −0.10 (0.01)*** | −0.07 (0.01)*** | 0.06 (0.01)*** | −0.01 (0.01) |
| Leisure words | −0.14 (0.01)*** | −0.09 (0.01)*** | 0.06 (0.01)*** | −0.05 (0.01)*** |

*Note*: Partial correlations control for general vocabulary, negative, and positive emotional tone. All tests are two-tailed. Coefficients are expressed as **r** (SE). For 95% confidence intervals and exact significance values, see Supplementary Table 7.
[a]For analyses involving age, $n = 9805$ authors' blogs.
[b]Coded 0 = male, 1 = female.
***$p < 0.001$.
**$p < 0.01$.
*$p < 0.05$.
†$p < 0.10$.

controlling for pre-writing levels of these specific moods. These effects remained above and beyond effects of potential confounds. It appears that people use larger EVs to describe states they are likely to intensify. However, the student sample and stream-of-consciousness exercise limit generalizability. Although the topic was not constrained, the exercise required some minimal degree of introspection, which could have inflated EVs. Study 2 addressed these concerns by analyzing a larger and more heterogeneous sample of natural language.

In Study 2, Psychometrics of EVs generally replicated Study 1. Although people used positively valenced words more frequently than negative words, EVs were again slightly larger for negative emotions. Split-half reliability exceeded test-retest reliability found in Study 1, suggesting stability. EVs again correlated with established markers of attention to internal experience (cognitive processing), general vocabulary breadth, and emotional tone, suggesting construct validity. Notably, most people tended to have small EVs, averaging about seven unique emotion words per 1000 words in their blog entries. This low rate is consistent with the limited nature of active vocabularies relative to other knowledge levels[16]. Negative EV was again correlated with virtually all markers of lower adjustment. Bloggers with a larger negative EV used language in ways consistent with people who are depressed, socially and behaviorally withdrawn, and in poorer physical health. Similarly, and even more consistently than in Study 1, negative EV demonstrated incremental validity in predicting well-being indices above and beyond effects of general vocabulary and emotional tone. Relationships of positive EV with well-being indicators were more mixed and only partially replicated Study 1.

Across both studies, then, people who used a wider variety of negative emotion words appeared to be faring less well; they used linguistic markers of lower well-being and reported greater depression, neuroticism, and poorer physical health. Conversely, people who used a variety of positive emotion words appeared to be faring well; positive EV was associated with linguistic markers of well-being and, in the student sample, self-reports of higher conscientiousness, extraversion, agreeableness, and overall health, and lower depression and neuroticism. Most relationships could not be attributed to the emotional tone of the texts, nor to the size of individuals' general vocabularies, recommending EV as a unique psychological marker in its own right. The stability of EV indices, acceptable for measures of its kind[39], underscores this potential.

To interpret these findings, the relationship of EV to mood (Study 1) is instructive. Larger EVs corresponded with state mood and its intensification, suggesting that emotion vocabularies not only provide insight into frequently experienced emotions, but perhaps also indicate a sort of emotional expertise: the tendency to use reflective thought to intensify already-present feeling states. If vocabulary size indicates interest[26], larger EVs may reveal emotional states preoccupying the individual. Future studies incorporating trait preoccupation with moods (e.g., rumination) may be fruitful. EV's correspondence with mood is noteworthy given possible difficulties inferring momentary well-being from emotion word frequencies[40,41]. The EV approach bypasses this issue by relying not on frequency but rather on the diversity of emotion word categories. Future research could explore whether EVs develop over time in parallel with the frequency of felt experiences, which would help confirm whether EVs serve as observable markers of familiar emotional states.

The current pattern of findings suggests that the relative proliferation of emotion words in individuals' active vocabularies may correspond to emotional experiences, but it does not speak to whether EVs were instrumental (helpful or harmful) in bringing emotional experiences about. Even though moods intensified during the stream of consciousness writing (Study 1) corresponding to diversity in emotion word use, the absence of experimental manipulation makes it impossible to conclude whether broader EVs *caused* this intensification. At the same time, it is also not possible to rule out such a causal effect. Language facilitates mental processes and subtly alters experience[31,33]. For instance, verbalizing taste sensations aids later memory retrieval[42], suggesting words may sustain fleeting subjective states. Emotion labels, in particular, may influence which emotions individuals perceive in others and experience themselves[32,34,36,43]. Future research should investigate

experimentally whether the state mood intensification such as we observed could have been constructed in part by more elaborative use of emotion synonyms. If so, this could be interpreted in line with constructivist emotion theories: while finding a precise emotion labels is presumed to aid in emotion regulation because it creates access to relevant emotion knowledge[5,44], applying more than one relevant label may be counterproductive for downregulating negative states, given that the labels may also reify the perception and felt experience of the state being named[34].

It is tempting to use the current findings to speculate about whether and how broad emotion vocabularies may be psychologically adaptive (i.e., functions causally to increase individuals' well-being). While we believe the current findings are a small part of a larger puzzle on this topic, we caution readers against interpreting current findings strongly in terms of the psychological adaptiveness of large emotion vocabularies. We take as given the existing larger framework, at the intersection of evolutionary psychology and appraisal theories, holding that the existence of language is advantageous because of its ability to segment and categorize experience into cognitively manageable units[45–47]. Recent discovery of cross-cultural universals in the structure of emotion semantics underscores the species-wide advantageousness of lexical systems containing names for both positive and negative states[48]. That said, evolutionarily advantageous behaviors do not necessarily confer advantage in social–emotional life at individual difference level[49]. A true test of psychological adaptiveness of large emotion vocabularies still requires experimental evidence that this study does not provide.

Importantly, our findings are not incompatible with evidence of regulatory neurological effects of labeling vs. not labeling negative states[14]. Perhaps unhappy people use larger vocabularies for negative emotions as attempts to down-regulate those states— maybe even somewhat successfully. Negative mood intensified as a function of negative EVs (Study 1); however, mood increases might have been even more pronounced had unhappy individuals not deployed an arsenal of emotion words. The findings presented here are silent on the possible coping function of emotion labels; rather, they complement existing literature by advancing the possibility that EVs serve as trait-like indicators of familiar states. Additionally, note that individual people likely rely on active and passive EV for different purposes in their everyday lives. Passive EVs are likely important for recognizing and interpreting day-to-day social behaviors in the social environment[50]. Active EVs may be more important for decoding one's own mental state and communicating it to oneself and others, to address one's own needs and inform new behavior[51,52].

Limitations of this study include the use of a one-item question to assess self-reported physical health and the absence of a direct measure of writers' education levels. A possible proxy for education, the general vocabulary index, was correlated with the breadth of emotion vocabularies, especially positive EV. However, many associations between EV and well-being indicators survived the inclusion of general vocabulary size as a covariate. Future replications with nuanced measures of individual differences in both well-being outcomes and intellectual functioning are needed. Our current findings also suggest several new research questions that the present study was not equipped to answer fully. For one, supplemental analyses using complementary text analysis methods suggest that emotion vocabularies may have interesting relationships with more sophisticated topic modeling approaches (Supplementary Note 2, Supplementary Table 4). Relationships between EV and conceptually related traits, such as emotional intelligence, also need to be articulated, as do relationships of EV dimensions to one another. Post hoc moderation analyses were suggestive of a possible buffering effect of rich positive emotion

vocabularies, such that they may protect against maladaptive correlates of rich negative emotion vocabularies with depression (Supplementary Note 1). This initial cross-sectional effect requires replication and extension before it can be substantively interpreted.

Several features of the EV method require comment. It is important to note that the words captured by the EV index are not applied to mental or emotional life exclusively; for instance, the word "alone" does not always refer to the feeling state. This is a natural feature of word-counting approaches to text analysis. Because words correspond to mental content imperfectly, text-derived signals are necessarily noisy; even robust, meaningful effects are necessarily small[53]. Given the explicit instruction in the stream-of-consciousness essays to focus on internal experience, it is likelier that words with multiple meanings were used for their emotional meaning more in this sample than in the blogs. As with other word counting approaches, the multiple meanings of words do not invalidate the EV approach, but instead simply constrain the inferences it can support. As argued elsewhere, spontaneously used words should be considered not explicit, but rather semi-implicit indices of thematic content, concerns, and frequent mental operations[54,55]. Moreover, the reliability of inferences from such data is proportional to sample size—this approach is best suited to revealing sample-wide patterns; inferences about any individual or small group should be treated with skepticism. See Supplementary Note 3 for additional nuance regarding the negligible impact of imbalance and length in EV word lists.

Future longitudinal research can ultimately determine whether and how changes in emotion vocabularies and changes in well-being are related. Intensive multilevel studies of emotion language are needed that would be capable of unpacking the complex relationship of emotion words to moods at contrasting temporal resolutions. Perhaps rich vocabularies for negative emotions are co-activated during distressed states so as to appear correlated in the short term, while in the long term these rich vocabularies could nevertheless be helpful. Future large-scale studies could further characterize the relationship of EVs to clinical presentations and changes in clinical functioning over time. Given the distinct conceptual viewpoints afforded by different measures, it would be interesting to include in such studies other measures related to emotion awareness, including traits[56], emotional abilities[11,57], and passive emotion word knowledge and recognition[12,13,44]. Understanding patterning of EV development within a multi-method, multivariate context will be essential to improve theoretical models of emotion language and support emerging emotion labeling-based interventions. To complement these observational methods for emotion language research, there is also a clear need for experimental manipulations of EV and other aspects of naturalistic, active emotion language generation. For example, a recent experimental manipulation of emotion labeling suggests preliminarily that leading individuals to generate excessive numbers of emotion labels in the context of a simulated stressor could undermine problem solving and emotion regulation efforts[20]. Many more experiments are needed before we can answer causal questions on the effects of emotion language on emotion experience and psychological adaptation.

At this juncture clinical application of the present methods and present findings is premature. Assessments of EV could potentially merit consideration as the basis for a future tool for planning treatment and predicting patient responses to affect labeling and other emotion-focused interventions. However, given the current method's more appropriate use for describing sample-wide patterns, the use of EV for diagnostic and intervention purposes would require further validation. While cross-sectional, our findings would be consistent with the possibility that distressed people may not need to increase vocabulary size, per se, for articulating their unhappiness—their negative EVs appear

larger already. Perhaps other features of emotion language, or coordinated habits or skills, are needed instead of—or in addition to—emotion language diversity. Instead of or in addition to EVs, psychological effects of emotion language may hinge on several other factors, including perhaps: context-specificity/precision of emotion language selections[18,44], deep conceptual emotion knowledge[34], cognitive efficiency of emotion naming processes[13,20], and/or accompanying mental stance (e.g., non-reactiveness[58]; nonjudgmentalness[59]; perceived clarity of emotions; and/or ambiguity tolerance[10,60]). Because words may help construct experience[34], we cautiously speculate, based on our positive EV findings, that positive EV may be an especially fruitful target for mechanistic and applied study.

Overall, the current project highlights the potential value of big data in the multi-method study of emotion language, because it can reveal broad patterning of naturally-occurring social and clinical processes. Large data sets make visible relatively small effects that are difficult to capture in lab samples. Our initial replication suggests that we are detecting small, but reliable, phenomena at the intersection of emotion vocabularies, distress, and well-being. This project also offers up a computerized tool for the quantification of active emotion vocabularies in participant-generated natural speech/text, which can aid the efficiency and generalizability of future efforts to understand the link between active emotion vocabularies and experience.

## Methods
### Study 1

*Participants and procedure.* Undergraduates enrolled in a large online introductory psychology class completed identical writing assignments in mid-September (Time 1) and, for test–retest reliability analysis, again in early December (Time 2). Most essays met criteria for inclusion: of the 1579 students who wrote the first essay, 1567 produced analyzable texts (i.e., at least 100 words; at least 70% of words identifiable by the default LIWC[38] lexicon); 1360 of those produced analyzable essays at Time 2. Given the novelty of our research question, effect sizes were not available a priori for a power analysis. However, this sample size is sufficient to detect even very small-to-moderate correlations with adequate (0.80) power. Students completing the Time 1 essay had a mean age of 18.8 (SD = 2.0), and 60.7% identified as female. Time 2 essay completers differed from non-completers by sex (females more likely to complete the second essay) and by reporting higher conscientiousness (independent *t* tests, *p* < 0.01). The procedure was approved by the Institutional Review Board at the University of Texas at Austin. Informed consent was obtained from all participants at the start of the semester.

*Stream of consciousness essays.* Participants recorded their thoughts in writing as they occurred for 20 min. Students completed the exercises on personal computers outside of class. The instructions read:

*During this 20-min task, your goal is to track your thoughts, perceptions, and feelings as they occur to you. Simply write continuously for the entire time and type out your thoughts, perceptions, and feelings without censoring them. There are no right or wrong things to write. Just track what is going on in your mind for the full 20 min.*

When 20 min had elapsed, students were given the option to stop or continue writing. Essays averaged 665 words (SD = 241) at Time 1 and 631 words (SD = 254) at Time 2. Sample essays representative of the tone/content of essays and range of EV scores appear in Supplementary Methods.

*Emotion vocabulary (EV).* LIWC[38] is frequency-based, meaning that it counts the number of occurrences of over 4500 words and word stems in over 70 categories. However, in its typical application, LIWC would produce the same score for texts containing ten different emotion words as it would for texts repeating the same emotion word ten times. We created an approach that quantifies the size of EVs by counting the rate of unique, or non-repeated, emotion words. By emotion words, we mean words that are used primarily for naming emotional states or feelings (e.g., happy, disheartened, and embarrassed), rather than referring to affectively tinged or themed content (e.g., victory, idiotic, and fight). While there is disagreement about the number of distinct emotional states[61], this approach is flexible and can be modified to include words of interest to each researcher. For the current project, the words naming positive and negative emotions were identified from the initial set of 406 positive and 499 negative affectively tinged words in the LIWC2007 emotion dictionaries. The final list of emotion words included 92 negative emotion and 53 positive emotion words (Supplementary Table 1). Normative data on age-of-acquisition (AoA) drawn from Kuperman and colleagues[62] show comparable AoA for positive (8.21 yr, SD = 2.63) and

negative words (8.9 yr, SD = 2.79; Supplementary Table 3). Note that the length and balance of final word lists can be expected to impact results negligibly, thanks to properties of count-based text analyses (Supplementary Note 3). By unique and non-repeated, we mean that the EV approach is concerned with the diversity—not frequency, quality, or other dimension—of emotion word use. Importantly, various inflections of the same word (e.g., sad, sadness, sadly) were counted as the same emotion word.

To eliminate the confounding effect of text length, all EV scores controlled for word count using the following formula:

$$\text{Emotion vocabulary (EV)} = \left(\frac{\text{\# Unique emotion words}}{\text{Total word count}}\right) \times 100.$$

Thus, EV scores represent the number of unique emotion words as a percentage rate of total word count. For example, the text, "he was so angry at me, but sadly there was nothing I could do" would receive an EV score of 2/14 × 100, or 14.29. Scores were computed separately for negative and positive emotion words (in the numerator position). To anticipate more fine-grained questions related to state mood, EV rates were also computed separately for names of sadness-related emotions (e.g., disappointed, bitter, and hopeless), anxiety- or fear-related emotions (e.g., nervous, afraid, and alarmed), and anger-related emotions (e.g., mad, furious, and aggravated). To correspond to state mood ratings for a stressed mood, which is generally considered an undifferentiated negative emotional state, the EV rate was also computed for general, negatively valenced words that could easily refer to an affective state (e.g., awful, terrible, and bad).

We have developed a free, open-source software program called Vocabulate, which automatically performs the text processing method described here. The software itself and dictionary file from the Supplemental Online Materials are available at https://osf.io/8ckyp/. This open source repository also contains the data supporting the findings of both studies reported in this paper.

*Individual difference indicators—text-derived.* Several other language markers were computed in order to better understand the EV construct. The first of these was computed using our open-source software for computing EV (Vocabulate; see "Code availability"). The rest were computed using LIWC, which counts words in approximately 80 categories that have been extensively validated in psychological research[25]. Categories are grammatical (e.g. articles and pronouns), thematic (e.g., social and religion), and psychological (e.g., positive and negative affect words). LIWC produces scores reflecting the presence of words in each category as a percentage of each individual's total word count.

*General vocabulary size (via Vocabulate).* EV size might be an artifact of general verbal ability or educational background. To address this possibility, we estimated each writer's general vocabulary size, which is a commonly used proxy for education level[63], as a type/token ratio (TTR). In calculating TTR, the number of unique words (types) is divided by the number of total words (tokens) used in the text. Like EV, TTR is expressed as a percentage, with higher values representing higher diversity in vocabulary. In this context, a unique word refers to any word that appears at least once in a given text. For example, the exclamation "A horse! A horse! My Kingdom for a horse!" contains five unique words out of nine total words (a, horse, my, kingdom, for). In order to capture the most relevant index of general vocabulary, only open-class, or content, words were counted (i.e., excluding function words such as pronouns, prepositions, articles, and other short and common words which are used frequently but are not clearly linked with verbal ability). To avoid redundancy, all emotion words used to compute EV were excluded. The average TTR for general vocabularies was 71.01, (SD = 7.25; range: 30.73–96.38).

*Cognitive processing (via LIWC).* Given that people may develop more expansive vocabularies to describe topics they find interesting, EV size was expected to converge with the tendency to reflect on internal experience. The LIWC cognitive process index, which captures the frequency of words such as think, question, and because. This index is believed to indicate individuals' efforts to analyze or mentally organize experience[64].

*Emotional tone (via LIWC).* One would assume that negative EV size might be highly correlated with the overall emotional tone of the text. To explore this possibility, negative and positive emotional tone was calculated for all texts using the LIWC negemo and posemo variables, which have been demonstrated in many studies to accurately reflect affective traits and states[65,66].

*Language diagnostic of well-being (via LIWC).* LIWC includes several non-emotion language dimensions that have been related to mental and physical health[25]. The category health includes 294 health-related words (e.g., clinic, flu, and pill). People who use more words in this category tend to report being less healthy than people who do not. Several studies have found that the use of first-person singular pronouns, or I-words, is correlated with depression, physical illness, anxiety, and even suicide[54]. Conversely, consistent with the social support literature, the more people use words suggesting engagement with others, such as first-person plural, or we-words, the fewer health problems they report[24,25]. In addition, words related to affiliation (e.g., ally, friend, and social),

achievement (e.g., win, success, and better), and leisure (e.g., cook, chat, and movie) were presumed to be related to higher psychosocial adjustment.

*Individual difference indicators—self-reported.* Self-report measures administered over the course of the semester were used to examine the individual differences associated with EV and confirm the utility of linguistic proxies for well-being.

*Personality.* Students completed the 44-item Five Factor Inventory[67] midway through the semester. The five factors include extraversion, neuroticism, agreeableness, conscientiousness, and openness. For the current sample, the internal reliability was good ($\alpha$s range from 0.78 for conscientiousness to 0.85 for extraversion).

*Physical health.* During the second week of the semester, participants responded to the question, "Overall, how would you rate your health?" Responses ranged from 1 (poor) to 5 (excellent). The mean response was 3.69 (SD = 0.84).

*Emotional health.* Three weeks before the semester ended and the Time 2 essay was completed, students completed the short form of the Center for Epidemiological Studies Depression Scale, which was developed for use in the general population (CESD-10)[68]. Participants rate the frequency with which they experienced 10 depression symptoms in past week on a scale ranging from 0 (rarely or none of the time/less than 1 day) to 3 (most or all of the time/5–7 days). In our sample internal consistency was good ($\alpha$ = 0.85). The mean depression score was 10.00 (SD = 5.91).

*State-level mood ratings.* Immediately before and after the 20-min writing exercise, students rated how much they felt four negative moods (sad, worried, angry, and stressed) and four positive moods (happy, enthusiastic, optimistic, and calm) on a Likert scale ranging from 1 (not at all) to 5 (a great deal). Ratings of the same valence were averaged to create negative and positive mood scores at both pre- and post-writing with acceptable-to-good internal consistency ($\alpha$s 0.75–0.83).

## Study 2

*Text corpus and author characteristics.* The complete text from 37,296 blogs was collected from blogger.com in August 2004. For complete information on corpus design, see Schler et al.[69]. Blogs contained all entries written from their inception to the day they were collected. Inclusion criteria were identical to that reported in Study 1, and duplicate texts collected in error were removed. The final corpus contained the full content of blogs by 35,385 individuals, ranging in total length from 107 to 481,983 words (M = 3142; SD = 6572). This sample size mitigates potential issues due to small effect sizes for work of this nature. Bloggers' self-identified gender and age were collected when available. Age and gender data were available for 27.4% of bloggers (N = 9688). Among these, approximately half of bloggers were female (5048; 52%), and ages ranged from 13 to 88 (M = 22.41; SD = 8.06). No informed consent was obtained, as identifying data were not collected.

*Emotion vocabulary.* Emotion vocabulary (EV) was computed using the same method described in Study 1. Representative examples of emotion words captured appear in Supplementary Methods.

*Individual difference indices (text-derived).* Because self-reports were not available, we relied on the text-based indices of well-being identified in Study 1. Evidence of criterion validity of these text-derived indices, based on their correlations with self-report in Study 1, is available online (Supplementary Table 2). Other text-based indices (cognitive processing, general vocabulary, positive and negative emotional tone, and presence of well-being themes) were derived for all blogs in the corpus using the same methods described for Study 1. The average TTR for general vocabularies was 62.91 (SD = 13.14, range: 4.66–95.06).

**Reporting summary**. Further information on research design is available in the Nature Research Reporting Summary linked to this article.

## Data availability

The datasets generated during and analyzed during the current studies are available in The Open Science Framework repository, at https://osf.io/8ckyp/. A reporting summary for this Article is available as a Supplementary Information file. Source data are provided as a Source Data file. Source data are provided with this paper.

## Code availability

The mathematical formulas for computing EV and related indices are provided in this paper, and the list of word mappings are provided in the online supplements. The custom open-source software we have developed to perform this computation directly, Vocabulate, is available at https://osf.io/8ckyp/ and https://github.com/ryanboyd/Vocabulate.

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

## Acknowledgements

We are indebted to Jason Ferrell, who collected and organized the data for Study 1. Preparation of this paper was aided by grants from the National Science Foundation (IIS-1344257, PI: Mihalcea), the Templeton Foundation (48503 and 61156, PI: Mihalcea), and National Institute of Mental Health (T32 MH018951, PI: Brent; R01 GM112697, PI: De Choudhury). The views, opinions, and/or findings contained in this report are those of the authors, and do not reflect the position, policy, or decision of any funding agency unless so designated by other documents.

## Author contributions

V.V. and J.W.P. developed the concept and approach for these studies using archival data collected by J.W.P. V.V., R.L.B., and J.W.P. developed the emotion vocabulary dictionary. R.L.B. parsed text samples to generate linguistic data, and V.V. conducted the statistical analyses. V.V. drafted the paper, with critical revisions provided by R.L.B. and J.W.P.

## Competing interests

The text analysis program LIWC is a commercial product co-owned by J.W.P. Proceeds from his share of the profits are all donated to the University of Texas at Austin. The authors declare no competing interests.
