## [Peer Review File · Nature Communications]

Reviewers' comments:

Reviewer #1 (Remarks to the Author):

This paper presents two very interesting studies examining the relationship of using a rich/varied emotional vocabulary to psychological functioning. The studies and findings are highly novel. Conceptually, the paper examines the possibility that emotion words are acquired to explain experiences. As such, those with more experience of sorrow, distress, etc. should have richer emotional vocabularies than others. The results support this idea in an innovative way--by examining the actual use of unique positive and negative emotion words in stream of consciousness writing and in an enormous sample of blog posts. It is just very interesting that emotional states give rise to emotion words and then contribute to those moods, in turn. Although I have some suggestions and concerns, overall I found this work to be fascinating.

I had a couple of suggestions that might improve this report. First, I wondered about the joint contributions of positive and negative unique emotion vocabularies (either additively or in interaction) in predicting the mental health outcomes described in Study 1. Although the clarity of correlations and partial correlations is a lovely aspect of the paper, it would be interesting to know whether there is a potential buffering effect of highly differentiated positive emotion vocabulary that might moderate the association between negative emotion vocabulary with distress. Are mixed emotions good? bad? Neither?

For Study 2, I was confused by the presentation of the N for the study. Were the blogs written by the same underlying sample of people? I assume that the 35,000+ refers to people (who wrote blogs of varying lengths) but it is not clear in the way it is presented. I think the source of confusion is that the term "unique blogs" sounds more like blog posts rather than whole blogs (which is what I think the authors did). This should be clarified.

The authors do not note that generally emotion researchers have long discussed the fact that the English language has more words for negative states than positive states. I wondered if they have considered the implications of this for their findings.

I think the Discussion might incorporate a bit more about the difference between emotion vocabularies that might be measured using something like an emotional intelligence measure or an emotion vocabulary test) and the data they have here. It would be interesting to probe the link between emotional experience and the expression of emotion in writing in the context of such information. Are those who are not distressed lacking these words or do they simply have no reason to deploy them in the moment? It might be interesting for future research to employ mood inductions to help illuminate the nature of rich emotion language. It seems there may still be a functional role of these rich vocabularies. If a person is experiencing distress, might it be better in a long term way, for the person to have names for these experiences? Over the long haul, it may be that those who are able to label their feelings with precision are better off than others.

This is a signed review by Laura King

Reviewer #2 (Remarks to the Author):

This paper offers an interesting, novel approach to investigating how emotional vocabularies used in natural language can predict mental and physical health, with Study 2 supplementing Study 1's findings with a larger sample and wider range of written texts. In light of the innovativeness of its emotional vocabulary (EV) methodology, this work is potentially highly impactful and may jumpstart a new line of research harnessing big data to identify markers of health and well-being (and ill-being).

To improve the paper, I list a few mostly minor questions below (not in order of importance).

1) The introduction and literature review of this paper seemed a bit disjointed and did not have ideal flow. At times the main ideas were not clear and were somewhat confusing, especially when transitioning between previous findings about active and passive vocabularies. For example, the authors include text about indigenous hunter-gatherers in the Philippines that does not appear to be relevant for the rationale of the study.

2) What was the purpose of students in Study 1 completing two essays? Was it just to compute test-retest reliability? It was unclear how the two sets of essays were used or treated differently in analyses.

3) Was it possible that students in the class used in Study 1, who completed all kinds of measures (as well as the two essays) over the course of the semester, could have guessed or become aware of the aims of the study? What was the cover story used? Was any material relevant to the study (emotion, emotion differentiation, LIWC, natural language processing) covered in the course?

2) When did participants complete the depression measure in Study 1? Also, as the authors are aware, the one-item global measure of physical health is not ideal. Thus, interpretations relevant to health (as opposed to well-being/mental health) should be taken with caution.

3) For text-derived individual differences, the authors account for general vocabulary size, but do not assess or mention participants' education level for either study. This may influence their findings, especially with the significantly wider age range in Study 2 (ages 13 to 48).

4) The writing samples for Study 1 included in the supplemental materials are very helpful and improve understanding of the findings. It would be beneficial to include classification information for each sample. Specifically, the authors could note or rank which samples have a larger negative emotion vocabulary relative to their positive emotion vocabulary.

5) The authors list a formula used to assess EV while controlling for total word count. A very similar formula is used for analyzing general vocabulary size. It would be helpful to elaborate a little on the difference between “unique words” and “unique emotion words.” The authors address what classifies unique emotion words, but do not list the inclusion criteria for unique words.

Reviewer #3 (Remarks to the Author):

In principle, we know little about natural emotion word repertoires or spontaneously produced emotion language in real world situations, and that the distinction between active and passive vocabularies is of particular value in this regard. That being said, the manuscript appears to be framed in an either-or sort of fashion which rings false. The overall impression is that the work is trying to overturn existing findings rather than contextualize them – and in doing this overstates its own conclusions. Even the title is misleading given the findings: it is mainly diverse vocabularies for negative emotions that appear to be linked to self-reported negative outcomes (i.e., ‘markers of distress’ are not objective, whereas the work being criticized measured outcomes in a more objective fashion). In addition, the authors own findings suggest that diverse vocabularies for positive emotions are linked to positive aspects of self-reported well-being.

Beyond this change in framing, there are several concerns with regard to method and interpretation that should be addressed in a substantive revision:

Method:

1. The authors do not provide any details on how emotion words were selected for use in their custom dictionary. Were these based on previous literature, normative ratings, corpus data (e.g., frequency of use)? Many of the words included (e.g., alone, bad, bitter) are not solely applied to mental life or emotional experience. Further, it seems possible that the extreme imbalance between the number of negative and positive emotion words may have artificially inflated negative EV scores relative to positive EV scores, casting doubt on whether the two indices can be directly compared with each other and with other text-derived and self-report variables. I would like to see the authors redo their analyses with a comparable number of emotion words in each index, and provide evidence of principled inclusion criteria.

2. The use of a coarse-grained, lexico-centric means of linguistic analysis is limited in the insights it can provide. More nuanced features of the language used in relation to emotion may be uncovered by a deeper analysis of themes, such as those attainable through topic modeling and distributional semantics. While the authors mention that there was a wide range of thematic content, this diversity is not quantified in any way. What does a thematic analysis suggest, and how do the present results relate to common themes identified? The relationships between the EV indices and other text-derived measures seem like obvious illustrations (e.g., negative emotion words are linked to health issues, positive words to achievement).

3. The choice of criterion validity measures in Study 1, and the lack of criterion validity measures in Study 2, diminishes the impact and utility of these findings. In Study 1, text-derived measures are not linked to robust measures that are necessary to fully contextualize the present findings. For example, it seems possible that responses to a single-item self-report measure of physical health may be influenced by current mood in a way that longer-form measures or certainly objective measures would not be. While the authors discuss the possibility that words influence our experience of the world (i.e., a moderate linguistic relativity hypothesis), they do not sufficiently demonstrate that language has any bearing on non-linguistic variables. Without any non-text-derived measures of mood, health, and well-being, a critical review of Study 2 results in particular would suggest they merely demonstrate the effect of language on language.

Interpretation:

4. The logic linking the effect and importance of words/concepts for emotion is not clear at even seems contradictory. To what extent do the authors hold that words can be adaptive in their segmenting of emotional experience? Given the current findings, is this only applicable to positive experiences, and what is the mechanism underlying that distinction? I would like to see the authors specifically state their alternative hypotheses for how language may (or may not) support emotional health and well-being.

5. Word usage (and therefore active vocabulary) isn't just a matter of individual 'comfort' or 'interest': there are complex processes underlying both lexical selection (within the individual brain) as well as communities of speech (between individual brains). Factors such as recency and frequency, prestige and affiliation, are also involved in the creation and maintenance of word repertoires. The authors' assertion that people who are suffering from distress are more interested in negative emotion words (p. 9, line 157) should be discussed along these lines.

6. The authors criticize previous findings without offering clear recommendations for how emotion language should be acquired and used. They suggest that it is not vocabulary size that needs to be increased, but stop short of proposing other features of language or the conceptual system that may be driving positive outcomes. Existing lines of research do investigate these mechanisms, yet are criticized by the authors for employing constrained/passive methods. This seems to me an opportunity to acknowledge that a multi-method approach may be necessary to fully investigate the underlying relationship between emotional experience and language use.

Miscellaneous:

7. Can the authors indicate how the sample texts in the supplemental material were selected? It is not clear if they are intended as representative cases, outliers, or random samples.

8. Scatterplots of key correlations would be useful so the reader can examine the distributions.

Reviewers' comments:

Reviewer #1:

General Comments: This paper presents two very interesting studies examining the relationship of using a rich/varied emotional vocabulary to psychological functioning. The studies and findings are highly novel. Conceptually, the paper examines the possibility that emotion words are acquired to explain experiences. As such, those with more experience of sorrow, distress, etc. should have richer emotional vocabularies than others. The results support this idea in an innovative way--by examining the actual use of unique positive and negative emotion words in stream of consciousness writing and in an enormous sample of blog posts. It is just very interesting that emotional states give rise to emotion words and then contribute to those moods, in turn. Although I have some suggestions and concerns, overall I found this work to be fascinating.

Comment #1: I had a couple of suggestions that might improve this report. First, I wondered about the joint contributions of positive and negative unique emotion vocabularies (either additively or in interaction) in predicting the mental health outcomes described in Study 1. Although the clarity of correlations and partial correlations is a lovely aspect of the paper, it would be interesting to know whether there is a potential buffering effect of highly differentiated positive emotion vocabulary that might moderate the association between negative emotion vocabulary with distress. Are mixed emotions good? bad? Neither?

Author reply: At the reviewer's suggestion, we have explored this idea with additional moderation analyses in the Study 1 sample. Using a multiple regression technique, we examined the effect of the interaction of Positive and Negative EV predicting self-reported depression scores, controlling for both EV main effects and the three covariates used in the manuscript's partial correlations (negative and positive emotional tone and general vocabulary). Results revealed a significant moderation suggestive of a buffering effect of positive emotion vocabularies, $b = -.84$, $SE = .40$, $p = .035$, with this interaction accounting for 3.4% of depression variance. As Figure S5 shows, the pattern of effects was such that for individuals with small negative emotion word repertoires, depression symptoms were lower and did not depend on Positive EVs. By contrast, for individuals with large Negative EVs, there was a buffering effect of Positive EV, such that individuals with restricted Positive EV were more depressed. Interestingly, however, this buffering effect reached significance only at exceedingly high levels of Negative EV (i.e., above values of 1.27, or above the 97.3rd percentile).

The reviewer wonders whether mixed emotions are good or bad. Given the meaning of the EV index, we cannot draw conclusions about mixed emotional states per se—only about co-occurring broad vocabularies. Based on the moderation findings, we cautiously suggest that possessing and/or using a varied positive emotion vocabulary mitigates the relationship between varied negative vocabularies and depression, but we cannot comment on the causality in this effect. Given the preliminary nature of these findings, their subtlety, and the reviewer's comment favoring the clarity of the correlational approach of the manuscript, we present these results in the Supplemental materials (S5), and we direct the reader here in Footnote 4 to the Study 1 results (p. 14).

Comment #2: For Study 2, I was confused by the presentation of the N for the study. Were the blogs written by the same underlying sample of people? I assume that the 35,000+ refers to people (who wrote blogs of varying lengths) but it is not clear in the way it is presented. I think the source of confusion is that the term "unique blogs" sounds more like blog posts rather than whole blogs (which is what I think the authors did). This should be clarified.

Author reply: We have clarified "unique blogs" to indicate that the N of 35,000+ refers to whole blogs (i.e., all content ever posted by 35,000+ individual people, each to their own personal blog), not separate blog posts. The revised sentence now reads: "The final corpus contained the full content of blogs by 35,385 individuals, ranging in total length from 107 to 481,983 words (M = 3,142; SD = 6,572)."

Comment #3: The authors do not note that generally emotion researchers have long discussed the fact that the English language has more words for negative states than positive states. I wondered if they have considered the implications of this for their findings.

Author reply: We thank the reviewer for this suggestion, which is also relevant to ideas raised by Reviewer 3. We have incorporated a discussion of this imbalance in negative and positive emotion words in the lexicon (pp. 24-25).

Comment #4: I think the Discussion might incorporate a bit more about the difference between emotion vocabularies that might be measured using something like an emotional intelligence measure or an emotion vocabulary test) and the data they have here. It would be interesting to probe the link between emotional experience and the expression of emotion in writing in the context of such information. Are those who are not distressed lacking these words or do they simply have no reason to deploy them in the moment? It might be interesting for future research to employ mood inductions to help illuminate the nature of rich emotion language. It seems there may still be a functional role of these rich vocabularies. If a person is experiencing distress, might it be better in a long term way, for the person to have names for these experiences? Over the long haul, it may be that those who are able to label their feelings with precision are better off than others.

Author reply: Thank you for sharing these speculations, which we found very interesting. We have elaborated the manuscript's Discussion in several ways throughout, inspired directly by these reviewer comments. We now directly discuss the need for future experimental work to better understand the nature of emotional vocabulary diversity and its complex relationship to mood and experience.

Reviewer #2:

General Comments: This paper offers an interesting, novel approach to investigating how emotional vocabularies used in natural language can predict mental and physical health, with Study 2 supplementing Study 1's findings with a larger sample and wider range of written texts. In light of the innovativeness of its emotional vocabulary (EV) methodology, this work is potentially highly impactful and may jumpstart a new line of research harnessing big data to

identify markers of health and well-being (and ill-being).

To improve the paper, I list a few mostly minor questions below (not in order of importance).

Comment #1: The introduction and literature review of this paper seemed a bit disjointed and did not have ideal flow. At times the main ideas were not clear and were somewhat confusing, especially when transitioning between previous findings about active and passive vocabularies. For example, the authors include text about indigenous hunter-gatherers in the Philippines that does not appear to be relevant for the rationale of the study.

Author reply: Thank you, we agree there were ways in which the flow of ideas in the introduction could have been clearer, and we have made changes to the introduction to address this issue that were informed by comments from each reviewer.

Comment #2: What was the purpose of students in Study 1 completing two essays? Was it just to compute test-retest reliability? It was unclear how the two sets of essays were used or treated differently in analyses.

Author reply: Yes, the second essay was used to compute test-retest reliability in Study 1. We have clarified this in the manuscript by replacing the term “temporal stability” with “test-retest reliability” in the introduction to Study 1 (p. 6), and by revising the paragraph in the Study 1 procedure so that it links the Time 2 essay explicitly to its function for these analyses: “Undergraduates enrolled in a large online introductory psychology class completed identical writing assignments in mid-September (Time 1) and, for test-retest reliability analysis, again in early December (Time 2).” (p. 7).

Comment #3: Was it possible that students in the class used in Study 1, who completed all kinds of measures (as well as the two essays) over the course of the semester, could have guessed or become aware of the aims of the study? What was the cover story used? Was any material relevant to the study (emotion, emotion differentiation, LIWC, natural language processing) covered in the course?

Author reply: It is not possible that the students guessed or became aware of the aims of this study, as the present study was conducted archivally; the research question was only conceptualized in 2015 after the data were collected in 2014. Questionnaires were completed to introduce students to topics as they were being taught in the course, such as issues related to self-report methodologies and cognition. Essays were initially collected as part of the introductory psychology course material to give students an opportunity to learn about mind-wandering, consciousness, and attention and gain an appreciation of William James’s stream of consciousness ideas. Topics related to emotion and language were covered during the course of the semester as well. To address the broader possibility that students may have anticipated their language would be analyzed, we have now read, at random, 80 (5%) of the Time 1 essays. Of those, 19 essays (24%) made some reference to possible readers and/or use in research (e.g., “Is anyone even going to read this?IS ANYONE OUT THERE????”; “Sorry I’m making this so long, whoever has to read this probably isn’t having a brilliant time;” “well anyways i hope this is some help to researchers because i actually tried”). Notably, only one text of the 80

(0.01%) gave any indication that the writer thought word choice might be analyzed (i.e., “I don't know what the purpose of this assignment is, but maybe it'll be scanned for keywords or something.”). Given this low rate, the archival nature of the study, and the fact that students were never introduced to the idea of emotional vocabulary diversity, we believe that students' expectations almost certainly could not have biased study findings.

Comment #4: When did participants complete the depression measure in Study 1? Also, as the authors are aware, the one-item global measure of physical health is not ideal. Thus, interpretations relevant to health (as opposed to well-being/mental health) should be taken with caution.

Author reply: The depression measure was administered on November 12, 2015, which was about 3 weeks before the end of the course (and about 2-3 weeks before the Time 2 essay used for test-retest reliability). We have added a caveat regarding the interpretive caution required for the one-item health measure to the manuscript (manuscript p. 23).

Comment #5: For text-derived individual differences, the authors account for general vocabulary size, but do not assess or mention participants' education level for either study. This may influence their findings, especially with the significantly wider age range in Study 2 (ages 13 to 48).

Author reply: We also thought that emotion vocabularies could have been affected by education level, and this was the motivation behind the creation of the general vocabulary index we tested as a covariate. We have now clarified in the manuscript that the general vocabulary index was intended to measure general verbal ability, a widely-used proxy for education level (see, e.g., Keuleers, Stevens, Mander, & Brysbaert, 2015), and we discuss the limitations related to absence of an education variable (p. 23).

In the Study 1 (college) sample, the education levels of participants were highly homogenous: 99.2% were undergraduates (59.7% freshman; 26.1% sophomores). For the blog sample, educational data was not available.

To shed further light on the possible role of educational attainment on the observed effects, we now report the average Age of Acquisition (AoA) for each word included in the EV computations (norms are in the new Supplemental Table S6, and we refer to them in the manuscript in Footnote 2). This AoA data was obtained from a published corpus of AoA norms (Kuperman, Stadthagen-Gonzalez, & Brysbaert, 2012). This reputable database of AoAs reflect receptive knowledge of 30,000 high-frequency English words. These AoA norms support the idea that the words in our EV dictionary are overwhelmingly learned in childhood or adolescence and are familiar to most native speakers by an average age of 9 years old (average AoA for negative words 8.9yrs, SD=2.79yrs; average AoA for positive words 8.21yrs, SD=2.63 yrs). Thus, it is likely that most (i.e., virtually all) of the college sample (Study 1) and broader-aged sample (Study 2) knew the dictionary-contained words and was able to use them with understanding and intent.

Comment #6: The writing samples for Study 1 included in the supplemental materials are very

helpful and improve understanding of the findings. It would be beneficial to include classification information for each sample. Specifically, the authors could note or rank which samples have a larger negative emotion vocabulary relative to their positive emotion vocabulary.

Author reply: We are glad that the samples (we understand this to mean in Supplement S1) were helpful in illustrating both the concept of active EVs and the general patterns in our results. In S1, each sample now appears together with the writer's EV scores and corresponding percentile ranks. We have also taken the opportunity to improve the interpretability of sample words captured in Supplement S3, where we have added EV scores to allow readers several options for how to the examples—including how negative emotion vocabulary compare relative to their positive emotion vocabulary within samples. Because word count is also indicated, readers can grasp by example how the EV scores adjusted for word count. To improve the interpretability of the EV scores in S3, we also now repeat here the sample's range and central tendency information, for greater context.

Comment #7: The authors list a formula used to assess EV while controlling for total word count. A very similar formula is used for analyzing general vocabulary size. It would be helpful to elaborate a little on the difference between “unique words” and “unique emotion words.” The authors address what classifies unique emotion words, but do not list the inclusion criteria for unique words.

Author reply: Thank you for pointing out this opportunity for greater clarity. We now state in the manuscript (p. 10) that “unique words” refers to the number of words that appear at least once in any given text. We additionally provide an example sentence that illustrates the concept of what constitutes a “unique word” as well as explicit description of what classes of words were included in these formulae (e.g., exclusion of function words, which are not diagnostic of vocabulary abilities).

Reviewer #3:

Comment #1 (Framing): In principle, we know little about natural emotion word repertoires or spontaneously produced emotion language in real world situations, and that the distinction between active and passive vocabularies is of particular value in this regard. That being said, the manuscript appears to be framed in an either-or sort of fashion which rings false. The overall impression is that the work is trying to overturn existing findings rather than contextualize them – and in doing this overstates its own conclusions. Even the title is misleading given the findings: it is mainly diverse vocabularies for negative emotions that appear to be linked to self-reported negative outcomes (i.e., ‘markers of distress’ are not objective, whereas the work being criticized measured outcomes in a more objective fashion). In addition, the authors own findings suggest that diverse vocabularies for positive emotions are linked to positive aspects of self-reported well-being.

Author reply: We agree that one of the primary contributions of the current research is to shed light on how active, unprompted emotion vocabularies behave, as well as their psychological relevance. Given the general lack of research on this topic, we also agree that there is considerable value in expanding our current understanding of emotions and affective language,

building on the extensive research that has been conducted on passive emotion vocabularies. In our revision, we have made it clear that we are not making an “either-or” distinction, and we have used our revisions to the Discussion (detailed in other replies above and below) to contextualize our findings within past work. We also acknowledge that humans possess and make use of active and passive vocabularies, likely in different ways (as our findings suggest), and that our current findings are not inherently incongruent with past work found in the emotion term recognition/labeling literatures. Instead, we now explicitly call attention to the fact that active EVs constitute another side of the coin, which has seen little to no empirical, psychological research to date (pp. 22-23). To improve the alignment between the title and study findings, we have also changed the manuscript title.

Comment #2 (Method): The authors do not provide any details on how emotion words were selected for use in their custom dictionary. Were these based on previous literature, normative ratings, corpus data (e.g., frequency of use)? Many of the words included (e.g., alone, bad, bitter) are not solely applied to mental life or emotional experience. Further, it seems possible that the extreme imbalance between the number of negative and positive emotion words may have artificially inflated negative EV scores relative to positive EV scores, casting doubt on whether the two indices can be directly compared with each other and with other text-derived and self-report variables. I would like to see the authors redo their analyses with a comparable number of emotion words in each index, and provide evidence of principled inclusion criteria.

We now include a thorough discussion of these concerns in the manuscript, including the selection of the words and the concern that words are not applied solely to mental life or emotional experience (see pp. 24). Regarding the imbalance between the number of negative and positive emotion words in the emotion vocabulary calculations, the reviewer raises an interesting and complex issue, which we answer in three parts:

- 1. This imbalance is to be expected, based on the long-observed fact that the English language has more words for negative states than positive states (as Reviewer 1 brought up in Comment #3). Moreover, it appears that the imbalance between negative and positive words in the lexicon may transcend cultures/languages (Shrauf & Sanchez, 2004). Although we agree that balanced word lists are intuitively appealing, artificially constraining our word lists would result in a methodological approach that is at odds with the normative sociolinguistic and psycholinguistic realities that have been empirically established across languages and cultures. We now comment explicitly on this imbalance that exists across languages and its implications for the measurement of emotion vocabulary and present findings (pp. 24-25; see reply to Reviewer 1, Comment #3).*
- 2. We agree in principle that, for the purpose of measuring vocabulary, a tool should have a ceiling high enough to detect variability in higher ranges. In practice, however, it is well established that sophisticated or advanced words for positive emotional states are rarely used in natural speech (see Bednarek, 2008). The imbalance in the length of word lists would thus not result in quantitatively meaningful differences. This is due to the nature of count-derived linguistic analytic approaches: having a larger candidate list of potential words that could appear from one category does not inherently translate to more words from that category actually appearing in spontaneous use. This principle is foundational to language-based domains of research, including corpus linguistics, which have*

established that the overwhelming majority of possible words that could be used to convey any specific concept are almost infinitesimally rare (Piantadosi, 2014; Zipf, 1935, 1949). To highlight this principle in our own samples, we provide a few examples here. In the student sample (Study 1), we find that the majority of words counted (74%) were used by fewer than 1% of students, and 25% were only used once in the entire corpus. The low impact of adding words to the lists is also evidenced by the new additions to our word lists (which we made during the course of this revision; see the following response point), which affected scoring for very few individuals. For instance, the added word “enthralled” occurred in < 0.5% of the blogs (Study 2) and not at all in the student writing (Study 1). (By comparison “love,” “happy,” and “excited” occurred, respectively, in 65%, 47%, and 22% of blogs, and 46%, 35%, and 24% of student texts.) Thus, although lengthening the positive word list raises the ceiling for possible positive EV scores, this affects a particularly small minority of individuals, and therefore does not alter the descriptive, sample-wide characteristics and relationships discussed in our work. In this way, word-counting-based approaches, while they would be imprecise for individual-level diagnostic assessment, are robust against measurement noise at the group level. We now comment on the negligible effects of word list length and balance, in the supplement (S8), and underscore the appropriate uses of the EV approach related to this and other reasons in the Discussion (pp 25-26).

3. Lastly, to address this issue as thoroughly as possible, we revisited the word lists to improve the balance as much as possible. We added 13 more positive words to the positive word list, re-ran all study analyses, and fully updated the manuscript. The updated word list has replaced the previous version found in Supplement S1 and is also included in the free *Vocabulate* software available for reader download. As a result of this change, the maximum possible Positive EV score was increased for each participant, and the possible mean Positive EV increased accordingly. However, as expected, these additions resulted in only small descriptive changes, and not at all to the broader patterns of results as observed through a variety of indicators. Old and revised positive EV scores correlated with each other at $r=.97$ (Study 1) and $r=.98$ (Study 2), and Positive EV scores increased for fewer than 23% of students (Study 1, Time 1) and for 23% of blog writers (Study 2). The magnitudes of any changes were extremely small: on average, positive EV increased in Study 1 by .04 points (SD increase = .09), and in Study 2 by .02 points (SD increase = .01). The largest increase in either study was for a student whose unique positive emotion count went up by 1 word, from 4 to 5 unique words; her positive EV score went up accordingly from 2.86 to 3.57 (a .71 difference), although she retained her relative rank in the sample as having the second highest positive EV. Most importantly, the primary study outcomes (correlation coefficients) involving positive EV shifted by a maximum of .02 in Study 1 and .02 in Study 2, resulting in no changes to the patterns of results.

Comment #3 (Method): The use of a coarse-grained, lexico-centric means of linguistic analysis is limited in the insights it can provide. More nuanced features of the language used in relation to emotion may be uncovered by a deeper analysis of themes, such as those attainable through topic modeling and distributional semantics. While the authors mention that there was a wide range of thematic content, this diversity is not quantified in any way. What does a thematic analysis suggest, and how do the present results relate to common themes identified? The relationships

between the EV indices and other text-derived measures seem like obvious illustrations (e.g., negative emotion words are linked to health issues, positive words to achievement).

Author reply: We agree that thematic analysis is an interesting avenue for developing an initial sense of contexts that may elicit EV variations, and we have thus added extensive analyses on thematic dimensions of the texts to the Supplements (new section S7). Principally, we used the meaning extraction method (MEM) topic modeling approach (see, e.g., Argamon, Koppel, Pennebaker, & Schler, 2007; Boyd, 2017; Chung & Pennebaker, 2008) to extract and quantify topics from each study in a data-driven manner. Unlike the LIWC categories used in the present study, which are determined a priori, MEM themes are derived in a bottom-up fashion from the emergent language patterns of the texts. By reducing the semantic dimensionality of words used in each corpus, we established and subsequently quantified the overarching “themes” or “topics” present in the writing samples from each study, respectively, which could then be statistically examined for their relationships to participant EVs.

To explore the reviewer’s question, we evaluated the correlations between MEM topic scores and EV scores. As the tables in S1 show, there were clear patterns of correlation, such that broader EVs were generally correlated in interesting and often intuitive ways with various topics. For example, in Study 1, students who more prominently invoked the topic of college (characterized by high loadings of the words “year,” “campus,” “college,” “degree,” “student,” “major”) used less diverse negative EVs ($r=-.08$, $p=.001$) and more diverse positive EVs ($r=.15$, $p<.001$). Students invoking the topic of sleep (“day,” “early,” “exhaust,” “late,” “hour,” “nap,” “bed”) used less diverse positive EVs ($r=-.13$, $p<.001$). In Study 2, bloggers who more frequently showed a poetic theme (“heart,” “soul,” “eye,” “tear,” “darkness,” “deep,” “light,” “dream,” “sky”) used more diverse emotion vocabularies of both valences, while bloggers who wrote more on the theme of recipes (“pepper,” “recipe,” “butter,” “salad,” “tomato,” “cut,” “stir”) used less diverse EVs of both valences. These results echo the primary findings in the manuscript, in that they give an impression of diversity in emotion language mirroring concerns with themes germane to distress and wellbeing.

However, it is important to note that the MEM themes are corpus-specific and, therefore, we cannot definitively state which themes may be diagnostic of other objective measures of wellbeing. We therefore must necessarily consider them to have only tentative explanatory value in this context with respect to our primary research questions, unlike the LIWC categories which have been demonstrated to be valid, highly reliable indicators of wellbeing and physical health across hundreds (if not thousands) of previous empirical studies. More extensive analysis of the topical data is thus beyond the scope of the present manuscript, which strives to examine the relationship of emotion vocabulary breadth to distress/wellbeing. Nevertheless, we agree that more computationally sophisticated approaches such as the MEM and semantic vector models will be useful in future work for unpacking and establishing how different emotion words may show concept-specific associations with various topical domains. In our revised manuscript, we now comment on the potential future uses of topic modeling and distributional semantic approaches in the Discussion (p. 23) and direct readers to the supplemental MEM analyses (Supplement S7).

Comment #4 (Method): The choice of criterion validity measures in Study 1, and the lack of

criterion validity measures in Study 2, diminishes the impact and utility of these findings. In Study 1, text-derived measures are not linked to robust measures that are necessary to fully contextualize the present findings. For example, it seems possible that responses to a single-item self-report measure of physical health may be influenced by current mood in a way that longer-form measures or certainly objective measures would not be. While the authors discuss the possibility that words influence our experience of the world (i.e., a moderate linguistic relativity hypothesis), they do not sufficiently demonstrate that language has any bearing on non-linguistic variables. Without any non-text-derived measures of mood, health, and well-being, a critical review of Study 2 results in particular would suggest they merely demonstrate the effect of language on language.

Author reply: We have made more explicit our acknowledgement of limitations related to the criterion measures in the current work (see response to Reviewer 2, Comment 4). Furthermore, to mitigate this limitation, we now provide additional, supplemental analyses that establish the validity of our text-derived individual difference variables and self-reported measures with regard to mental and physical wellbeing. Building on the data collected in Study 1, which contained both methods (Table S4, referenced in a footnote in Study 1 Results, as well as in Study 2 Method), we establish links between language-based measures of wellbeing and health, which are indeed consistent with the broader literature on linguistic cues and their diagnosticity of mental and physical health (see, e.g., Culotta, 2014; De Choudhury, Counts, & Horvitz, 2013).

Overall, the correlations between text-derived and self-reported indicators of well-being, which are reported online in Table S4, consistently indicate the associations suggestive of construct validity (i.e., I-words and illness words with low wellbeing; we-, affiliation, achievement, and leisure words with high wellbeing). Given that our findings are internally (i.e., across both studies) as well as externally consistent (i.e., converge with previous research), we are able to interpret the current results in a manner that supports the use of language-based measures as proxies of mental and physical wellbeing, acknowledging the limitations discussed above.

We also note that the self-report measure of physical health used in Study 1 was not collected concurrent to the first Stream of Consciousness writing sample – it is therefore not possible for the previous self-reported physical health measures to have been influenced by participant mood at the time of writing, which occurred at a later date.

Comment #5 (Interpretation): The logic linking the effect and importance of words/concepts for emotion is not clear at even seems contradictory. To what extent do the authors hold that words can be adaptive in their segmenting of emotional experience? Given the current findings, is this only applicable to positive experiences, and what is the mechanism underlying that distinction? I would like to see the authors specifically state their alternative hypotheses for how language may (or may not) support emotional health and well-being.

Author reply: This comment gives us the opportunity to clarify the scope of the current research—focused on the diversity of emotion vocabularies—in relation to more basic work on the adaptiveness of language/words in general, about which this reviewer seems to be asking. In doing so, we have clarified the manuscript to show that there is no contradiction here. Our

research question does not take issue with the established framework, at the intersection of evolutionary psychology and appraisal/categorization theories, which holds that the existence of language has been adaptive for the species because of its ability to segment experience into cognitively manageable units (Boster, 2017; Johnstone & Scherer, 2000). We agree with the existing framework that we are fitter as a species thanks to language, without which there would be only incommunicable, unsustainable “sensorimotor toil” that would ultimately interfere with reproduction (Cangelosi & Harnard, 2001). The segmentation of emotional experience—positive and negative—can be thought to operate along the same principles of adaptiveness as all forms of categorization in cognition (Harnard, 2017); i.e., humans would be disadvantaged by an absence of any language for positive and negative states. Recent findings of cross-cultural universals in the structure of emotion semantics underscore the fundamental species-level adaptiveness a lexical system for naming both positive and negative states (Jackson et al., 2019). That said, bringing to bear evolutionary-level knowledge on trait-level variability, while potentially useful, poses complex theoretical and methodological challenges not to be undertaken lightly, and which the current study is not designed to address (see Brown & Richerson, 2014; Buss 2015; Loehlin, 1992; Toobey & Cosmides, 1990). Our hypothesis here is that there will be a broad correspondence between vocabulary richness and experience. This is fundamentally concerned with patterning of individual differences in vocabulary size within species, which we now explain in the manuscript does not depend on the overall adaptiveness of language in general (pp. 21-22). In the process of making these changes, we have also added an elaboration to the discussion that places the current findings in deeper conversation with current constructivist theories of emotion language, and we show how current results and their implications may be consistent with constructivist theories in nuanced ways (pp. 21).

Comment #6 (Interpretation): Word usage (and therefore active vocabulary) isn’t just a matter of individual ‘comfort’ or ‘interest’: there are complex processes underlying both lexical selection (within the individual brain) as well as communities of speech (between individual brains). Factors such as recency and frequency, prestige and affiliation, are also involved in the creation and maintenance of word repertoires. The authors’ assertion that people who are suffering from distress are more interested in negative emotion words (p. 9, line 157) should be discussed along these lines.

Author reply: This is a good point. We did not intend to suggest that those individuals suffering from distress are more interested in negative emotion words. We have now clarified that, in accordance with cognitive and linguistic theories on vocabulary enrichment, the psychological process of being interested in or attentive to one’s own affective states can serve as a mechanism by which the individual is motivated to acquire and deploy more varied ways of describing their emotions. Additionally, we now explicitly incorporate past research that touches on these other psycholinguistic and sociolinguistic factors that are well-established and known to be involved in vocabulary development and maintenance: “In addition to well-established determinants of vocabulary acquisition and maintenance (e.g., Kenji & D’andrea, 1992; Van Overschelde, 2002), we similarly suggest that preoccupation with or interest in one’s own affective states could contribute to the development of increasingly diverse affective taxonomies and lexica.” (p. 5).

Comment #7 (Interpretation): The authors criticize previous findings without offering clear

recommendations for how emotion language should be acquired and used. They suggest that it is not vocabulary size that needs to be increased, but stop short of proposing other features of language or the conceptual system that may be driving positive outcomes. Existing lines of research do investigate these mechanisms, yet are criticized by the authors for employing constrained/passive methods. This seems to me an opportunity to acknowledge that a multi-method approach may be necessary to fully investigate the underlying relationship between emotional experience and language use.

Author reply: It is true, we had deliberately steered away from making explicit recommendations on how emotion language should be acquired and used, because the current correlational methods preclude such inferences. However, we share the reviewer's interest in this question, and we are familiar with the existing research that begins to speak to it. Upon reflection, we have decided that readers are likely to be wondering the same thing, and thus, we now explicitly discuss the temptation of—and reasons to refrain from--extrapolating from the current findings to form applied recommendations for acquiring/using emotion language. In doing so, we have named several other features that may drive positive outcomes (pp. 26-27).

Comment #8 (Miscellaneous): Can the authors indicate how the sample texts in the supplemental material were selected? It is not clear if they are intended as representative cases, outliers, or random samples.

Author reply: The sample texts were intended as representative combination of tone and content/concerns and a range of EV values. We now state this intention clearly in the manuscript (p. 7), and now also provide the sample central tendencies and ranges to complement the EV scores in Supplement S3 (per Reviewer 2, Comment #6).

Comment #9 (Miscellaneous): Scatterplots of key correlations would be useful so the reader can examine the distributions.

Author reply: We now provide scatterplots of key bivariate relationships in Supplement S9.

Additional changes made not in response to reviews:

1. We have improved the precision of two items in the negative emotion word list by differentiating “horrible” from “horror” and related forms, and “terrible” from “terror” and related forms. Changes in negative EV scores were negligible (they were unchanged for 99% of Study 1 participants and 93% of Study 2 participants). Correlation coefficients involving negative EV scores changed at most by .006 (Study 1) and .004 (Study 2).
2. A colleague has pointed out that, although positive EV could not be broken into discrete emotion families in the same way, we had missed an opportunity to conduct the state-level mood change analyses (Study 1) using the positive EV scores and positive mood ratings. Therefore, we have now added these analyses to the manuscript. Results essentially replicate the pattern observed for negative emotions, such that more

expansive vocabulary for a given emotional category was associated with increases in the corresponding mood.

***REVIEWERS' COMMENTS:

Reviewer #1 (Remarks to the Author):

I was positively disposed to this innovative work on the first submission and find that the authors have done a great job of responding to my concerns. I think this work is important and innovative and I do not have additional concerns.

Reviewer #2 (Remarks to the Author):

Title: FEELINGS IN TOO MANY WORDS: RICH EMOTION VOCABULARIES AS MARKERS OF DISTRESS

Manuscript #: NCOMMS-19-11846-T

I viewed the authors' revision and their detailed responses to each of my comments and concerns. I am impressed with the thoughtfulness and attentiveness to every face of my (and other reviewers') original critiques. I am persuaded that the results have potentially important implications for both theory and research (basic & applied). I wish the authors the best of luck with this manuscript and their future work exploring how emotional vocabularies might be used in natural language to predict markers of health and well-being.

Reviewer #3 (Remarks to the Author):

The authors have been very thorough and considerate in their response to my previous comments and concerns. The manuscript no longer reads as combative, and better illustrates how the current work complements existing approaches to understanding the complex relationship between language and emotional experience. This is a more nuanced and thoughtfully-interpreted piece that is clear about the value of its contribution. I also really like the new title and the supplementary analyses of thematic/topic content.

I do have a few smaller comments for additional revisions:

1. On page 16 (line 325), the authors write "It appears that people have larger EVs to describe familiar states." There is no direct evidence that these states are familiar; please rephrase.
2. On page 20 (lines 425-7), the authors write "Future research could explore whether EVs, more than emotion word frequencies, develop in conjunction with felt experiences and therefore serve as an emotional fingerprint of familiar states." Can the authors clarify their meaning here, or provide an example of how this might be accomplished? The authors could also incorporate a brief mention of work on the diversity of emotions in relation to biodiversity (emodiversity; Quoidbach et al, 2014; but see Brown & Coyne, 2017), which consider both evenness (i.e., frequency) in addition to sheer number (i.e., variety/range).

3. Points about the possible psychological adaptiveness of emotion language are raised at several points, and the discussion of emotion language in evolutionary terms seems like an unnecessary – or at least overly elaborate – divergence from the main narrative. I appreciate that these points were added in consideration of my previous comments, but believe they can be better integrated. Further, the authors seem to contradict themselves when they state on page 22 (lines 471-3) that their study does not attempt to provide evidence for the psychological adaptiveness of large emotion vocabularies. Indeed, the current work is correlational rather than experimental, but all the same suggests that larger positive EVs may be associated with beneficial outcomes. Perhaps it is to the exact mechanism (“label any given emotional experience most precisely”) that the authors wish to remain agnostic?

4. Miscellaneous: The LIWC acronym is used on p. 7 before it is introduced on p. 8. If possible, it would be helpful to introduce the state-level mood ratings before the corresponding emotion-specific EV scores are described. In Study 1, when did participants complete the CESD-10? Reviewer 2 asked this question previously, but the manuscript text was not updated. The scatter plots in the supplemental material need updating: the Study 2 positive EV figure is cut off, x-axis numbers do not need so many decimal points, and fit lines could be added to illustrate the reported effect(s).

Response to Reviewer Comments

As requested, we have copied and pasted all reviewer comments into this document verbatim.

Reviewer #1:

I was positively disposed to this innovative work on the first submission and find that the authors have done a great job of responding to my concerns. I think this work is important and innovative and I do not have additional concerns.

Reviewer #2:

I viewed the authors' revision and their detailed responses to each of my comments and concerns. I am impressed with the thoughtfulness and attentiveness to every face of my (and other reviewers') original critiques. I am persuaded that the results have potentially important implications for both theory and research (basic & applied). I wish the authors the best of luck with this manuscript and their future work exploring how emotional vocabularies might be used in natural language to predict markers of health and well-being.

Author Reply: We thank all the reviewers for their comments.

Reviewer #3:

The authors have been very thorough and considerate in their response to my previous comments and concerns. The manuscript no longer reads as combative, and better illustrates how the current work complements existing approaches to understanding the complex relationship between language and emotional experience. This is a more nuanced and thoughtfully-interpreted piece that is clear about the value of its contribution. I also really like the new title and the supplementary analyses of thematic/topic content.

I do have a few smaller comments for additional revisions:

1. On page 16 (line 325), the authors write "It appears that people have larger EVs to describe familiar states." There is no direct evidence that these states are familiar; please rephrase.

Author Reply: We have rephrased this language to improve accuracy. This sentence now reads, "It appears that people use larger EVs to describe states they are likely to intensify," which more closely reflects the direct evidence being discussed at that point in the manuscript.

2. On page 20 (lines 425-7), the authors write "Future research could explore whether EVs, more than emotion word frequencies, develop in conjunction with felt experiences and therefore serve as an emotional fingerprint of familiar states." Can the authors clarify their meaning here, or provide an example of how this might be accomplished? The authors could also incorporate a brief mention of work on the diversity of emotions in relation to biodiversity (emodiversity; Quoidbach et al, 2014; but see Brown & Coyne, 2017), which consider both evenness (i.e., frequency) in addition to sheer number (i.e., variety/range).

Author Reply: We have clarified the intended meaning of the sentence the reviewer quotes as follows: “Future research could explore whether EVs develop over time in parallel with the frequency of felt experiences, which would help confirm whether EVs serve as observable markers of familiar emotional states.” We also appreciate the reference to emodiversity and have incorporated a brief mention of this concept in Supplementary Note 3, which discusses the (negligible) impact of word list length on EV calculations. Specifically, the paper by Brown and Coyne (2017) provides a useful foil for our discussion in that note, because its critique of ceilings imposed by measurement tools in the computation of emodiversity was highly appropriate in that context, but is not relevant in ours. We did not incorporate the emodiversity construct into the main text, however, as the emodiversity construct does not directly pertain to emotion language.

3. Points about the possible psychological adaptiveness of emotion language are raised at several points, and the discussion of emotion language in evolutionary terms seems like an unnecessary – or at least overly elaborate – divergence from the main narrative. I appreciate that these points were added in consideration of my previous comments, but believe they can be better integrated. Further, the authors seem to contradict themselves when they state on page 22 (lines 471-3) that their study does not attempt to provide evidence for the psychological adaptiveness of large emotion vocabularies. Indeed, the current work is correlational rather than experimental, but all the same suggests that larger positive EVs may be associated with beneficial outcomes. Perhaps it is to the exact mechanism (“label any given emotional experience most precisely”) that the authors wish to remain agnostic?

Author Reply: We have better integrated—truncating by more than half—the discussion of emotion language in evolutionary terms that we had added in response to previous reviewer comments. We are grateful for the assistance in finding this balance. In making these changes, we have clarified the apparent contradiction to state more plainly that we view a true test of advantageousness of large emotion vocabularies (positive or negative) as requiring experimental evidence, which our study did not provide.

4. Miscellaneous: The LIWC acronym is used on p. 7 before it is introduced on p. 8. If possible, it would be helpful to introduce the state-level mood ratings before the corresponding emotion-specific EV scores are described. In Study 1, when did participants complete the CESD-10? Reviewer 2 asked this question previously, but the manuscript text was not updated. The scatter plots in the supplemental material need updating: the Study 2 positive EV figure is cut off, x-axis numbers do not need so many decimal points, and fit lines could be added to illustrate the reported effect(s).

Author Reply: We have addressed most of these miscellaneous comments. The LIWC acronym is introduced now on p. 7; the timing of the CESD-10 has been added to Study 1 Methods; scatter plots have been updated as requested. The only change we did not make was to move higher the introduction of the state mood ratings to a position ahead of the corresponding EV scores; we felt it was important to lead with the computation of the EV scores to maintain the conceptual focus on EV scores. We thank Reviewer 3 for their close attention throughout the review process.